# Redo Thyroidectomy: Updated Insights

**DOI:** 10.3390/jcm13185347

**Published:** 2024-09-10

**Authors:** Luminita Suveica, Oana-Claudia Sima, Mihai-Lucian Ciobica, Claudiu Nistor, Anca-Pati Cucu, Mihai Costachescu, Adrian Ciuche, Tiberiu Vasile Ioan Nistor, Mara Carsote

**Affiliations:** 1Department of Family Medicine, “Nicolae Testemiţanu” State University of Medicine and Pharmacy, 2004 Chisinau, Moldova; lumsuveica@yahoo.com; 2PhD Doctoral School of “Carol Davila”, University of Medicine and Pharmacy, 050474 Bucharest, Romania; oana-claudia.sima@drd.umfcd.ro (O.-C.S.); anca-pati.cucu@drd.umfcd.ro (A.-P.C.); mihaicostachescu@gmail.com (M.C.); 3Department of Clinical Endocrinology V, C.I. Parhon National Institute of Endocrinology, 011863 Bucharest, Romania; carsote_m@hotmail.com; 4Department of Internal Medicine and Gastroenterology, “Carol Davila” University of Medicine and Pharmacy, 020021 Bucharest, Romania; 5Department of Internal Medicine I and Rheumatology, “Dr. Carol Davila” Central Military University Emergency Hospital, 010825 Bucharest, Romania; 6Department 4—Cardio-Thoracic Pathology, Thoracic Surgery II Discipline, “Carol Davila” University of Medicine and Pharmacy, 050474 Bucharest, Romania; adrian.ciuche@umfcd.ro; 7Thoracic Surgery Department, “Dr. Carol Davila” Central Emergency University Military Hospital, 010825 Bucharest, Romania; 8Department of Radiology and Medical Imaging, Fundeni Clinical Institute, 022328 Bucharest, Romania; 9Medical Biochemistry Discipline, “Iuliu Hatieganu” University of Medicine and Pharmacy, 400347 Cluj-Napoca, Romania; tiberiu.nistor@umfcluj.ro; 10Department of Endocrinology, “Carol Davila” University of Medicine and Pharmacy, 050474 Bucharest, Romania

**Keywords:** thyroid nodule, ultrasound, surgery, thyroidectomy, thyroid cancer, redo, goitre, recurrent laryngeal nerve, parathyroid

## Abstract

The risk of post-operatory hypothyroidism and hypocalcaemia, along with recurrent laryngeal nerve injury, is lower following a less-than-total thyroidectomy; however, a previously unsuspected carcinoma or a disease progression might be detected after initial surgery, hence indicating re-intervention as mandatory (so-called “redo” surgery) with completion. This decision takes into consideration a multidisciplinary approach, but the surgical technique and the actual approach is entirely based on the skills and availability of the surgical team according to the standard protocols regarding a personalised decision. We aimed to introduce a review of the most recently published data, with respect to redo thyroid surgery. For the basis of the discussion, a novel vignette on point was introduced. This was a narrative review. We searched English-language papers according to the key search terms in different combinations such as “redo” and “thyroid”, alternatively “thyroidectomy” and “thyroid surgery”, across the PubMed database. Inclusion criteria were original articles. The timeframe of publication was between 1 January 2020 and 20 July 2024. Exclusion criteria were non-English papers, reviews, non-human studies, case reports or case series, exclusive data on parathyroid surgery, and cell line experiments. We identified ten studies across the five-year most recent window of PubMed searches that showed a heterogeneous spectrum of complications and applications of different surgeries with respect to redo interventions during thyroid removal (e.g., recurrent laryngeal nerve monitoring during surgery, other types of incision than cervicotomy, the use of parathyroid fluorescence, bleeding risk, etc.). Most studies addressing novel surgical perspectives focused on robotic-assisted re-intervention, and an expansion of this kind of studies is expected. Further studies and multifactorial models of assessment and risk prediction are necessary to decide, assess, and recommend redo interventions and the most adequate surgical techniques.

## 1. Introduction

One of the most challenging aspects of public health that physicians of primary and secondary health level should take into consideration involves the presence of the thyroid nodules, including a multinodular goitre (recently renamed “thyroid follicular nodular disease”) and, of course, potentially, an underlying thyroid malignancy. This condition (specifically, a thyroid nodule) is considered to be the most frequent endocrine pathology in the general population. Approximatively 1 to 5% of individuals suffer from at least one thyroid nodule across random neck ultrasound exams (mostly depending on the age and geographic area, etc.). However, since there is an age-dependent incidence, in certain adult categories, the prevalence of this ailment is up to 20 to 30%. Most of the thyroid nodules are not malignant, yet the rate of malignancy varies from 5 to 30% depending on the study’s criteria [1,2,3].

The most frequent types of thyroid cancer (i.e., the most frequent endocrine malignancies) are represented by differentiated forms in terms of papillary (most common) and follicular, which both account for 70–75% of all cases of thyroid neoplasia (originating from the follicular cell). Other types, such as medullary thyroid carcinoma (which is a neuroendocrine neoplasm originating from C para-follicular cells either in sporadic or hereditary syndromes) and anaplastic malignancy (which may be registered as a late anaplastic shift from differentiated thyroid cancers, also originating from the follicular cells), are more rarely found [4,5,6]. Nationwide iodination protocols that were applied in prior endemic countries during the last decades improved the panel of thyroid cancer in terms of registering a major shift from the follicular to the papillary type as being currently the most common histological form that offers better short- and long-term outcomes [7,8,9].

Additionally, multinodular goitre is still diagnosed, especially in previous endemic (iodine deficient) areas, and the associated panel of complications should also be analysed with regard to the endocrine, imagery (such as anterior neck ultrasound), and (if necessary) surgical management. In many cases, this starts with the primary care physician evaluation across various screening protocols [10,11,12]. Multinodular goitre may cause not only cosmetic complaints from a patient (adult or child) but it might also induce compressive symptoms/signs that impair breathing or deglutition, and sometimes even poses malignant risks [13,14,15,16]. In such cases, surgical intervention is the only therapeutic approach that relieves the symptoms, restores the patient’s quality of life, and ensures an overall good outcome [17,18,19]. This is performed as a total thyroidectomy (with or without lymph node removal), when the entire thyroid gland is removed or a lobectomy or hemi-thyroidectomy is performed (when part of the thyroid is left in place) according to the current guideline-based stratification and focused management [20,21,22,23]. While the risk of iatrogenic hypothyroidism and hypocalcaemia/hypoparathyroidism and recurrent laryngeal nerve injury is lower following a less-than-total thyroidectomy than a complete thyroid removal, a previously unsuspected carcinoma might be detected and, hence, indicates a mandatory re-intervention (so-called “redo” surgery) with thyroidectomy completion, which generally poses an elevated risk of complications when compared with the baseline surgical procedure [24,25]. The redo decision takes into consideration a multidisciplinary approach, but the surgical technique and the actual approach is entirely based on the skills of the surgical team (from the application of the standard protocols to personalised decisions) [26,27,28].

### Objective

We aim to introduce a review of the most recent data published with respect to redo thyroid surgery. As the basis of the discussion, a novel vignette on point is introduced.

## 2. Methods

This was a narrative review. We searched English-language papers according to the key search terms in different combinations such as “redo” and “thyroid”, alternatively “thyroidectomy” and “thyroid surgery”, across the PubMed database. Inclusion criteria were original articles (studies) that specifically provided an analysis of redo thyroid surgery from different perspectives (for example, recurrent laryngeal nerve monitoring during surgery, types of incision other than cervicotomy, the use of parathyroid fluorescence, bleeding risk, etc.). The timeframe for publications was between 1 January 2020 and 20 July 2024. Exclusion criteria were non-English papers, reviews, non-human studies, case reports or case series, exclusive data on parathyroid surgery, and cell line experiments (Figure 1).

Additionally, in the discussion section, a brief clinical vignette offers a practical basis of understanding and approach regarding the redo thyroidectomy from a multidisciplinary modern perspective. The parameters were introduced in terms of hormonal assays, antithyroid antibodies, serum tumour markers suggestive for thyroid malignancies, imaging assessments such as neck ultrasound and computed tomography, respectively, and post-operatory pathological reports. Also, we provided some intra-operatory captures from the redo thyroidectomy that was performed by a different surgical team from the first surgical procedure. Informed and written consent was obtained from the patient for the anonymous use of her medical data while being hospitalised.

## 3. Results: Sample-Focused Analysis in Redo Thyroid Surgery

### 3.1. Intra-Operative Neurophysiological Monitoring of the Laryngeal Nerves

The routine application of intra-operative nerve monitoring during a thyroidectomy still represents a challenging issue across daily surgical practice, and most modern data shows a heterogeneous use of this technique all over the world, with large variations between centres and even between surgeons coming from the same department [29,30,31]. Recurrent laryngeal nerve damage following thyroid removal still accounts for one of the most important complications (with a general rate of 3 to 6%, mostly 1.5 to 20%). Particularly, an elevated risk is detected in large goitres (although, among benign thyroid conditions, the highest risk is reported for Graves–Basedow’s disease) in suspected/confirmed malignancies and redo thyroidectomy; hence, a meticulous approach with respect to keeping the nerves’ integrity is essential [32,33].

Damage (either paresis or paralysis) to the recurrent laryngeal nerve represents one of the most dramatic hazards in the field of thyroidectomy regardless of the underlying histological condition. If the lesion is unilateral, hoarseness is displayed, while in bilateral involvement, dyspnoea and even acute respiratory insufficiency might develop, with various consequences from a reduced quality of life to a life threatening situation. An indirect laryngoscope exam performed before and after thyroid surgery specifically highlights the surgery impact, but the intra-operatory monitoring remains an essential key intra-operatory element [34,35,36].

With respect to this nerve injury, we mention two studies of different designs (survey type and retrospective analysis) that were identified based on our strategy of the PubMed search [34,36]. A nationwide survey study involving surgeons who had to deal with thyroid removal, such as head and neck oncologists (representing 55% of all 101 respondents) and general oncologists, along with endocrine surgeons and otolaryngologists, showed that 42.6% of them used intra-operative recurrent laryngeal nerve monitoring. A higher rate of applying it was found in those who had less than or equal to 15 years of surgical experience versus those with more than 15 years of experience (*p* = 0.02) for surgeons who associated with a total volume of >50 thyroid interventions versus <50 (*p* = 0.016). Interestingly, the main purpose of using this monitoring included medicolegal aspects and also surgeon comfort. Regarding the specific underlying pathological aspects, the most frequent use was for redo thyroidectomy (45.5%, meaning 46 out of the 101 responders) and in subjects who pre-operatively already displayed one fixed cord (38.6% of the responders performed neuromonitoring for this specific reason), while, in malignancies, the application was performed more often than in benign thyroid ailments (*p* = 0.016), as expected [34].

Moreover, a single-centre retrospective study evaluating the rate of recurrent laryngeal nerve damage on 255 consecutive patients (aged between 21 and 59 years, mean age of 39 years) revealed a unilateral lesion in 25/255 subjects (a total of 9.8% of the cohort). A unilateral vocal cord lesion affected 17/25 of them (meaning 6.7% of the entire cohort), and 3/7 of them, representing 1.2% of the initial cohort, showed a permanent injury. A bilateral vocal cord injury was detected in 8/255 participants (meaning 3.1% of the cohort) and all individuals from this sub-group presented a transitory pattern. The rate of the unilateral nerve injury was higher in total/near-total versus bilateral/unilateral subtotal thyroidectomy (25.9% versus 7.9%, *p* < 0.05), in malignant versus benign thyroid conditions (17.1% versus 4.7%, *p* < 0.05), and in men versus females (12.2% versus 8.8%, *p* < 0.05). Redo thyroidectomy (performed for recurrent goitre in 12 patients, representing 4.7%) was complicated by unilateral nerve damage (N = 1, 8.3%) and bilateral nerve damage (N = 2, 16.7%); in one patient (8.3%), the injury was permanent [36].

Of note, awareness is essential in redo thyroidectomy, since it brings this particular risk of unilateral recurrent laryngeal nerve palsy, despite using careful intra-operative neuromonitoring. Sometimes, achieving a positive response to the vagus nerve stimulation during surgery might not be enough and nerve palsy might occur anyway (as only seen and confirmed after the operation). Yet, an immobile vocal cord (with the preservation of its anatomical integrity) helps keep the voice [37].

### 3.2. Intra-Operatory Parathyroid Considerations during Thyroidectomy

Fluorescence-imaging techniques are based on the auto-fluorescence of the parathyroid glands (under near-infrared light) and angiography assessment (via applying fluorescent indocyanine green dye) of the parathyroid glands in order to avoid their lesions during a thyroidectomy or parathyroidectomy [38,39,40,41].

We identified two distinct studies to address the issue of parathyroid damage during thyroid surgery, especially in redo intervention, which is prone to a more complex panel of side effects [42,43]. One international Delphi survey-based study regarding the use of parathyroid fluorescence during a thyroidectomy showed the experts’ opinions as follows: auto-fluorescence is better than an indocyanine green angiography to localise the parathyroid but is inferior to pinpointing the parathyroid perfusion, while using indocyanine green is particularly important in redo surgery [42].

Moreover, we pinpoint a retrospective single-centre study on the use of a radionuclide occult lesion localisation procedure (0.5 mCi 99m-Technetium macro-aggregate albumin was injected into the lesion via ultrasound surveillance) during redo neck exploration in subjects (N = 25; female to male ratio of 21 to 4; mean age of 54.5 years) who were confirmed with a recurrent thyroid malignancy (N = 20 participants) or a parathyroid tumour (N = 5 subjects) between 2016 and 2018. The data on redo endocrine surgery revealed an average operative time of 100.6 min, and a rate of transitory hypoparathyroidism of 2/25 with a statistically significant reduction of the thyroglobulin levels versus pre-redo assays (*p* < 0.05) in individuals confirmed with a differentiated type of thyroid cancer [43].

### 3.3. Redo for Bleeding Complications

Nowadays, the risk of bleeding following endocrine surgery of the thyroid and parathyroid glands, while rare, still represents a very severe consequence, being considered a potential life-threatening emergency [44,45,46]. A retrospective 7-year cohort study on thyroid and parathyroid surgery complicated by post-operatory bleeding that required redo intervention revealed a 1.3% rate of the second intervention (25 patients out of the initial 1913 individuals). This rate of redo surgery was not correlated with the subjects’ age, gender, or surgical team, but it was higher following total versus hemi-thyroidectomy (*p* = 0.045) or parathyroidectomy (*p* = 0.001). Interestingly, the panel of factors that were not actually associated with the risk of bleeding was large and it included a redo intervention, neck dissection, drain use, or the administration of anticoagulants or the value of the body mass index, yet the risk of bleeding was correlated with a longer hospital stay (*p* = 0.001). The subjects who suffered from post-operatory bleeding had similar rates of wound infection and recurrent laryngeal nerve damage, along with hypoparathyroidism, when compared with those who did not show this potentially severe complication [47].

Moreover, since generally redo thyroidectomy is recognised to show a higher rate of bleeding than first-time intervention, we mention the results of a single-centre retrospective study (N = 4153 patients) on redo surgery without drains (between 1996 and 2018). A haematoma was diagnosed in 4% of those who underwent a redo thyroidectomy for completion or for recurrent thyroid ailment versus 5% in individuals undergoing a primary bilateral thyroidectomy, while the rate of transitory hypocalcaemia was 16% of 21% and 19%. The incidental unilateral recurrent laryngeal nerve paralysis was 7%, of 8% and 6%, with a similar rate concerning the rate of permanent nerve damage (1% for each of the three sub-groups) [48]. Similarly, a study on the UK Registry of Endocrine and Thyroid Surgery (between 2004 and 2018) showed a 1.2% rate of re-operation for post-operatory bleeding (0.9% following hemi-thyroidectomy) in 52,838 patients. Multivariable regression showed that the risk of bleeding was associated with male sex, advanced age, redo thyroidectomy, diagnosis of retrosternal goitre, and total thyroidectomy, but, overall, the authors concluded that haemorrhage risk could not be predicted [49]. Generally, the risk of developing a haematoma following thyroid surgery (regardless of being the first or second intervention) was shown to be associated with a certain predisposition of the patient, not the surgical techniques or surgeon’s skills [44,45,46,47,48,49].

### 3.4. Redo TORT (Trans-Oral Robotic Thyroidectomy)

TORT (trans-oral robotic thyroidectomy) stands for a new surgical perspective for thyroid ailments that require gland removal; however, the level of evidence is far from convincing in some areas, and we already know that only a sub-group of the patients diagnosed with thyroid conditions are actually TORT candidates [50,51,52]. Overall, the procedure seems safe and effective if the indication is clearly established, depending on the prior experience of the surgical team [53,54,55]. Despite obvious cosmetic advantages, redo TORT is yet a matter of debate concerning the overall benefits. To conclude, in this aspect, the current statistical evidence remains low [56,57,58].

According to our methods, we were able to identify a small sample size in this retrospective study conducted between 2017 and 2023 regarding redo TORT in order to offer a completion procedure. The cohort (N = 10 patients, female to male ratio of 7 to 3, average age of 42.2 years) was referred for the second TORT procedure either due to the progression of the initial aggressive tumour (5 of the 10 subjects) or due to the need of removing the remnant tissue, which was confirmed by a synchronous papillary thyroid cancer following initial TORT (5 of the 10 patients). The flat dissection time was similar between the initial and the redo TORT (*p* = 0.125); the median duration between the initial and the redo surgery was 6.5 months. Total operation time and console duration were higher in the first versus the redo surgery (*p* = 0.021 and *p* = 0.019, respectively). One-month transitory hypoparathyroidism was reported in two out of ten of the patients, with an overall post-redo surveillance of 21.5 months (and no recurrence was registered within this cohort) [59]. Further research is necessary to clearly highlight the role of TORT and its benefits for thyroid surgery candidates, especially in the redo approach.

### 3.5. Trans-Axillary Robotic Redo Thyroidectomy

As already mentioned, traditional (open or endoscopic) cervicotomy may be replaced by minimally invasive robotic surgery in certain circumstances, a novel area of surgical techniques that offers three types of access, namely trans-oral, trans-axillary, and bilateral axillo-breast approach [60,61,62]. The data we have so far are still debatable, but we expect a massive expansion within the following years, as similarly seen in other domains of robotic/minimally invasive thoracic surgery [63,64,65]. Noting this specific frame of robot-assisted thyroidectomy, we pinpoint a single-centre study on redo intervention for recurrent thyroid cancer (N = 65 participants with the trans-axillary robotic approach) between 2007 and 2021. Different procedures were performed, such as single completion total thyroidectomy (N = 26 subjects), modified radical neck node dissection (N = 16), and exclusive modified radical neck node dissection (N = 23). Most patients underwent redo intervention at the same site as the initial intervention and the most frequent histological type was papillary cancer. The rate of post-operatory complications was 26.2% (transitory hypocalcaemia), 4.6% (permanent hypocalcaemia), and 4.6% (injury of the recurrent laryngeal nerve), while structural post-redo recurrence was found in 1 of the 65 patients. Median post-operatory follow-up was 50.7 months [66].

To summarise, this current robotic trend, while not replacing the gold standard of open thyroid surgery, represents an emergent topic in the field of surgical technique development [67].

### 3.6. Cost-Analysis in Redo Surgery

As specified, the decision of redo surgery should be regarded from a multidisciplinary perspective, not only surgical, and from a cost-analysis point of view that might reveal a higher median dollars per minute in redo versus primary thyroidectomy (*p* < 0.0001), according to one study published in 2022 [68]. However, reimbursement protocols vary with each country and surgical centre; thus, a conclusion might not generally be applicable (Table 1).

## 4. Discussion

Re-operative thyroid surgery is not an uncommon procedure in the modern era. It is necessary in cases of persistent thyroid cancer or recurrent malignancy and in patients with multinodular goitre who have undergone a lobectomy; small foci of endocrine cancer were registered in a post-operatory pathological report [69,70,71,72]. Nevertheless, this type of re-intervention comes with greater technical challenges raised by the presence of fibrosis, oedema, tissue fragility, and a higher risk of local haemorrhage after the first procedure, according to most data (but not all) [73,74,75,76]. An increased risk of post-operatory complications is described (such as hypocalcaemia, injury of the recurrent laryngeal nerve, haematoma development, etc.), but the results are heterogeneous and they vary with the surgical technique, type of approach, type of gland resection, and the experience of the surgical team, etc. [77,78,79,80].

### 4.1. Clinical Vignette

#### 4.1.1. First Admission

For practical insights, we introduce a case where redo thyroidectomy was decided. This was a 41-year-old female patient referred by her primary physician for an endocrine check-up due to non-specific compression symptoms, such as recent dyspnoea and progressive dysphonia. The family medical history was irrelevant. On admission, the thyroid panel revealed a normal thyroid function in terms of TSH (thyroid stimulating hormone) of 1.3 μIU/mL (normal range between 0.5 and 4.5 μIU/mL), FT4 (free levothyroxine) of 11.3 pmol/L (normal range between 9 and 19 pmol/L), negative serum thyroid antibodies, namely anti-thyroperoxidase antibodies of 0.65 IU/mL (normal range between 0 and 5.61 IU/mL), and normal serum tumour marker calcitonin of 0.84 pg/mL (normal range between 5.17 and 9.82 pg/mL). Mineral metabolism assays revealed a total serum calcium level of 9.65 mg/dL (normal range between 8.4 and 10.3 mg/dL) and a parathormone level of 50 pg/mL (normal values between 15 and 65 pg/mL), thus excluding primary hyperparathyroidism, and adequate levels of 25-hydroxyvitamin D of 30.2 ng/mL (normal range between 20 and 100 ng/mL).

A neck ultrasound revealed an enlarged right thyroid lobe measuring 2 by 2 by 5 cm and a left lobe measuring 2 by 3 by 5 cm, with a hypoechoic multinodular pattern. The right lobe had a spongiform nodule of less than 1 cm and five other cysts with diameters between 0.3 and 0.7 cm were identified, while the left lobe also had a macro-nodule measuring 2.9 by 1.85 by 2.8 cm with a mixed (solid/cystic) structure and three other small cysts, with the largest diameter varying between 0.5 and 0.8 cm (Figure 2).

These findings were suggestive of a compressive multinodular goitre with euthyroidism that required surgical removal. An additional imaging procedure was performed due to compressive complaints. A computed tomography scan was also performed and detected another right paratracheal nodule in the superior mediastinum at the clavicular level (0.85 by 1.11 by 1.42 cm). This newly detected nodule was suspected to be a small ectopic thyroid tissue.

#### 4.1.2. Initial Thyroidectomy and Removal of the Ectopic Thyroid Tissue

The lady was referred to the thoracic surgery unit for a total thyroidectomy and ectopic tissue removal. Despite the pre-operatory multidisciplinary decision (by the endocrine surgeon) for total thyroidectomy, the intra-operatory decision shifted into a subtotal thyroidectomy in association with the resection of the ectopic thyroid remnants. The pathological report showed a right thyroid lobe having a papillary thyroid carcinoma and ectopic thyroid tissue next to the orthotopic gland (and no connective tissue) with benign features.

#### 4.1.3. Post-Operatory Assessments

One month following the surgical procedure, the patient came for an endocrine evaluation. The compressive symptoms completely remitted following the thyroid surgery. The clinical assessment was within the normal limits, except for a recent anterior cervical scar and local oedema following the surgical procedure. Blood assays revealed normal TSH of 4.06 μIU/mL (of note, the patient was not under any levothyroxine replacement, which seemed unnecessary on clinical grounds noting the incomplete thyroid removal). The serum post-operatory tumour marker thyroglobulin was high, 30 ng/mL (for a patient confirmed with differentiated thyroid carcinoma) and anti-thyroglobulin antibodies (22.35 IU/mL) were negative. A post-surgical neck ultrasound showed a remnant in the right thyroid lobe measuring 2 by 1 by 3 cm and in the left thyroid lobe measuring 2 by 1 by 3 cm, with a hypoechoic inhomogeneous pattern, a few micro-calcifications, and persistent postsurgical oedema. An intravenous contrast computer tomography scan revealed postsurgical antero-cervical fibrosis extended to the clavicular level measuring 6.68 by 2.02 by 2.37 cm (Figure 3).

Due to the remaining parts of both thyroid lobes, the thyroglobulin value could not be addressed as a tumour marker of persistent thyroid malignancy. The remaining thyroid tissue could not be destroyed via radioiodine ablation, noting its large diameter; thus, a redo surgery (with completion procedure) was mandatory.

#### 4.1.4. Redo Thyroidectomy

This time, the remaining thyroid was removed by a second surgical team with a rapid recovery and three-day hospitalisation duration. No side effects were registered, except for expected iatrogenic hypothyroidism that required levothyroxine initiation (Figure 4).

The pathological report revealed tissue fragments containing fibrous, connective, and adipose tissue, striated muscle fibres, thyroid tissue with hyperplastic areas, marked hyper-functional aspects, lymphocytic inflammatory infiltration, vascular congestion, haemorrhage areas, interstitial sclero-hyalinisation, granulomatous inflammation with multinucleate giant cells, and no other malignant aspects.

#### 4.1.5. Post-Redo Outcome

Two months following redo thyroidectomy, the patient came for an endocrine check-up. There were no complaints, and no clinical pathological finding was registered. Serum TSH levels were within normal ranges of 1.1 μIU/mL under levothyroxine replacement of 100 µg per day. Thyroglobulin significantly was decreased but remained elevated under replacement therapy of 3.9 ng/mL (she had negative anti-thyroglobulin antibodies of 20.75 IU/mL) (Figure 5).

A neck ultrasound showed no thyroid remnants with persistent post-operatory oedema, but no fibrosis was seen after the first procedure was diagnosed (Figure 6).

Following the stimulation with recombinant human TSH, serum thyroglobulin increased to 34.7 ng/mL. Whole body I^131^ radioiodine scintigraphy was performed, and it revealed remnant thyroid orthotopic tissue with bilateral radioiodine uptake; no other pathological uptake of the radioiodine tracer was identified. Considering these persistent high values, a dose of 30 mCi ^131^I radioiodine was administered. The subject will follow a lifelong plan of periodical check-ups that include TSH and FT4 assays, along with tumour marker thyroglobulin serial testing. The target TSH is ≤0.1 μIU/mL according to TSH suppressive levothyroxine therapy-based management. The potential negative issues of long-term treatment are elevated cardiovascular risk and osteoporosis (with associated increased fragility fracture risk), thus these aspects need to be taken into consideration in the long-term management [81,82,83,84].

Notably, the high values of thyroglobulin after redo thyroidectomy with the completion procedure confirmed the importance of the surgical completion in order to offer a better overall prognosis and to allow lower doses of radioiodine therapy. In this instance, despite the obvious intra-operatory challenges, no other post-surgery complication was noted. The extent of thyroidectomy that should be performed in low-risk differentiated thyroid carcinoma is still a subject of debate, as total thyroidectomy might be considered overtreatment, while partial/sub-total thyroidectomy comes with challenges regarding post-surgical follow-up in terms of serum thyroglobulin assay applicability if a large thyroid remnant is already present [85,86,87].

The post-operatory biomarker thyroglobulin represents a routinely performed analysis for differentiated thyroid cancer in cases with persistence or recurrence; however, after lobectomy/near-total/hemi-thyroidectomy, this marker has a limited use because it can be secreted either by the remaining cancerous cells or by the normal residual thyroid tissue [88,89,90]. This particular case study offers a practical perspective. The multidisciplinary decision of the redo intervention was performed since thyroglobulin levels were elevated and a large thyroid tissue was identified following the first surgery.

Another particular aspect was the presence of a rather large area of post-operatory fibrosis and even local oedema after the first and second surgical procedure—a rare complication that mostly depends on a person’ collagen particularities rather than the endocrine diagnosis itself or the surgical skills [91,92,93]. This aspect cannot actually be predicted in an individual who is referred for a thyroidectomy (unless a prior similar intervention was performed, as seen in this present case). Recently, artificial intelligence-based models have been released to predict the severity of the local collagen-associated issues, such as post-surgery scars, but these insights are still early results and are not part of daily practice in most countries [89].

Generally, local adherences and fibrosis are pitfalls in redo thyroidectomy and some experimental studies are under development to seek new formulas and plant extracts to counteract this unpleasant aspect [90]. Alternatively, pulse dye laser therapies have been used for hypertrophic scars upon thyroidectomy [91]. However, the impact on the overall quality of life with respect to post-operatory scars in patients who have been confirmed with thyroid cancer is low and an increased rate of satisfaction has been identified amid this condition, while most subjects associate with a minimal scar after one to six months since intervention [93,94]. As an alternative to open thyroid surgery, novel approaches have been introduced, as mentioned, with a rather low rate of adherence in most surgical centres so far, but further studies are expected to highlight their utility on a larger scale [95,96,97].

### 4.2. Current Limits and Further Expansion

These ten studies that we identified across a five-year most recent window of searches showed a heterogeneous spectrum of complications and applications of different types with respect to redo interventions during thyroid removal [34,37,42,43,47,48,49,59,66,68]. Whether the recent COVID-19 pandemic introduced a potential bias in referring (or delaying) the actual candidates to redo thyroidectomy might be taken into consideration, particularly during the first pandemic waves, since a shift in patients’ addressability and the underlying diagnoses has been observed in thyroid surgery, as seen in many other surgical domains [98,99,100].

It is mandatory to delve into the new WHO classification of thyroid neoplasms that was released in 2022, but the majority of the cited studies use traditional terms. At this moment, this latest classification is not generally applied in everyday cases. This novel nomenclature encourages new risk stratification strategies and, moving forward, we expect new insights regarding the placement of the redo thyroid surgery in relationship to the severity of the first diagnosis and, potentially, in association with new emergent biological markers [101,102]. Globally, the topic of thyroid malignancy embraces major importance due to the epidemiologic impact and increasing incidence amid modern medicine, being regarded as the sixth or seventh most common cancer in female patients [103,104]. We need new models of risk stratification, including not only emergent molecular and genetic factors but also elements of already-applied management such as prior thyroidectomy or previous use of radioiodine ablation [103,104,105].

Overall, a qualitative analysis with regard to the redo thyroidectomy includes several important chapters, e.g., the need for completion upon prior thyroid cancer confirmation (as seen in the case sample), the underlying disease progression (particularly in thyroid malignancies), the extemporaneous (histological) report of a thyroid neoplasia during a selective parathyroid tumour removal that further requires a thyroid surgery, the need for lymph node resection, and mandatory intervention of a post-operatory bleeding/haematoma following the first surgery [34,37,42,43,47,48,49,59,66,68,106].

Of note, we should include here another rare scenario of the “hidden”, “forgotten”, or “missed” thyroid gland upon initial thyroidectomy, namely an ectopic thyroid tissue, particularly at the mediastinal level that might complicate the lifelong follow-up of a differentiated thyroid cancer via using serum thyroglobulin assays, which remains high due to the ectopic thyroid and might mimic a thyroid cancer recurrence [107,108,109,110].

The key factors in redo thyroid surgery include the multifaceted decision of re-intervention, the most adequate surgical technique for re-operation, the risk of post-operatory complications and long-term outcome, and cost-analysis [34,37,42,43,47,48,49,59,66,68]. Whether a multi-layered risk calculator (a stratified decision) should include other elements such as genetic background (*RET* or *BRAF* pathogenic variants), metabolic interplay (e.g., the presence of obesity or diabetes mellitus), or the co-diagnosis of autoimmune conditions (such as autoimmune thyroid disease) or other coagulation anomalies represents a complex matter and an ongoing process [111,112,113,114,115,116]. We also need multidisciplinary models to assess the damage of recurrent laryngeal nerve and voice recovery, since multiple factors (other than the surgery itself) are involved [117]. Moreover, the risk of parathyroid involvement following redo surgery should be diminished by using, for instance, fluorescence-imaging techniques that are not limited to indocyanine green, which is a relatively new area. There are other ways to identify parathyroid glands such as “stress-tests”, 5-aminolevulenic acid, etc. [118].

Regarding limitations of the current work, we mention the narrative design via a single database search that allowed a more flexible approach, and the fact that we did not include articles reporting a good surgical outcome of the initial surgery, thus no redo thyroidectomy needed to be registered. Instead, we only focused on the articles that highlighted different data on redo surgery from complications to associated costs. Further longitudinal studies and models of risk prediction are necessary to indicate the need of redo intervention, taking into account the initial clinical, endocrine, and imaging assessment, the intra-operatory findings, the histological exam, post-surgery serum tumour, and molecular biomarkers, considering the ratio between the risks and benefits. Moreover, we need specific protocols to pinpoint the adequate type of redo surgery in terms of incision, extension of resection, and the use of robotic-assisted procedures and minimally invasive techniques by the surgical team.

## 5. Conclusions

We identified ten studies across a five-year most recent window of PubMed searches that showed a heterogeneous spectrum of complications and applications of different surgeries with respect to redo interventions during thyroid removal. Most studies addressing novel surgical perspectives focused on robotic-assisted re-intervention, and an expansion of this kind of study is expected. Further studies and the multifactorial models of assessment and risk prediction are necessary to decide, assess, and recommend a redo intervention and to indicate the most adequate surgical technique.

## Figures and Tables

**Figure 1 jcm-13-05347-f001:**
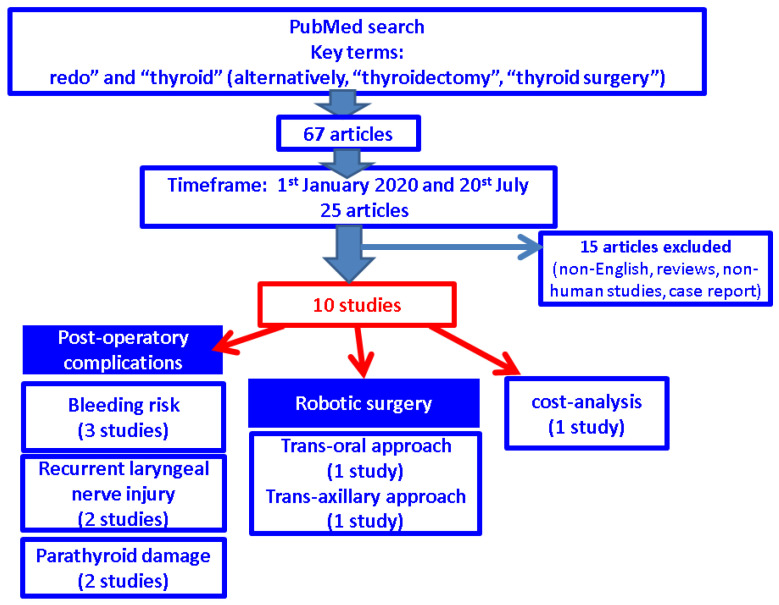
Flow chart of search according to our methods.

**Figure 2 jcm-13-05347-f002:**
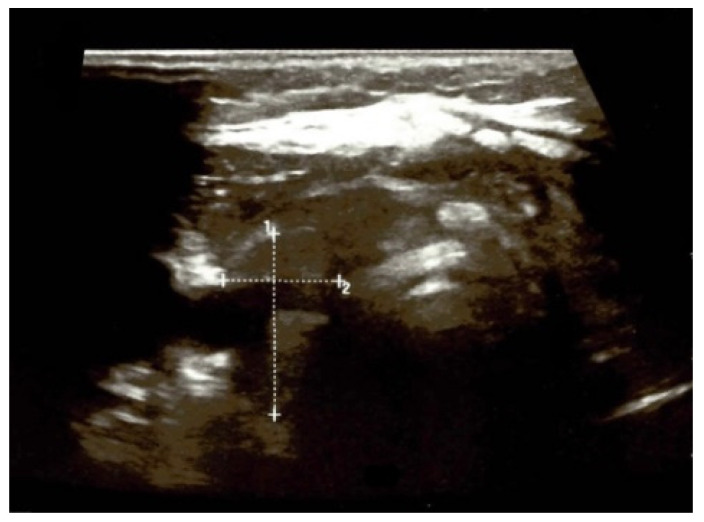
Anterior neck ultrasound on first admission of a 41-year-old female with multinodular goitre and compressive symptoms (the capture highlights the largest nodule on the left thyroid lobe).

**Figure 3 jcm-13-05347-f003:**
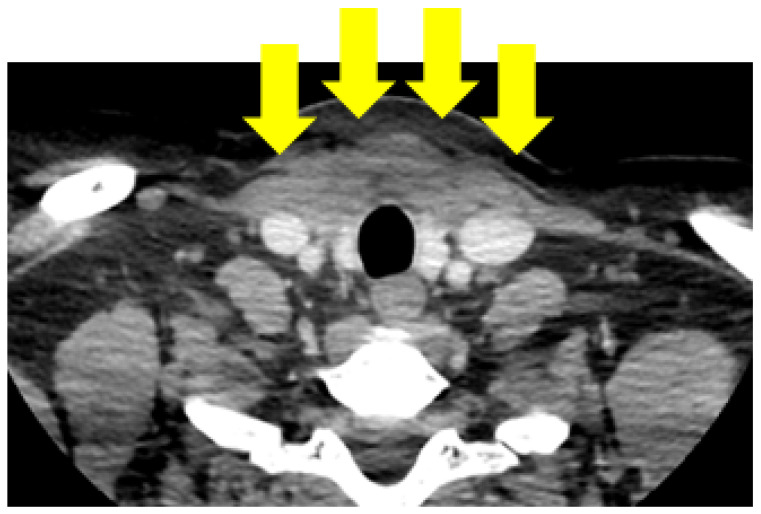
Computer tomography one month after first thyroid surgery showing post-operatory fibrosis and oedema (yellow arrows); an anterior neck capture (transversal plane).

**Figure 4 jcm-13-05347-f004:**
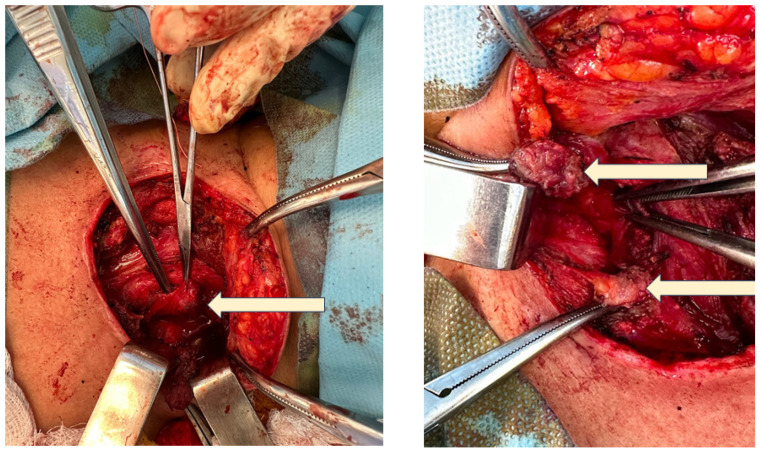
Intra-operatory aspects: (**upper left**) left thyroid remnant tissue (white arrow); (**upper right**) fibrosis following the first surgery (white arrows); (**lower left**) identification of the intact right superior parathyroid gland (white arrow); (**lower right**) left thyroid remnant (post-operatory specimen).

**Figure 5 jcm-13-05347-f005:**
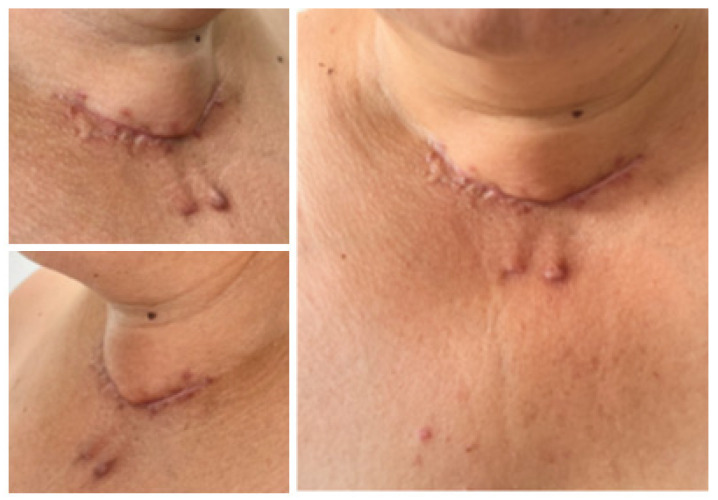
Skin aspect following redo thyroidectomy after 2 months.

**Figure 6 jcm-13-05347-f006:**
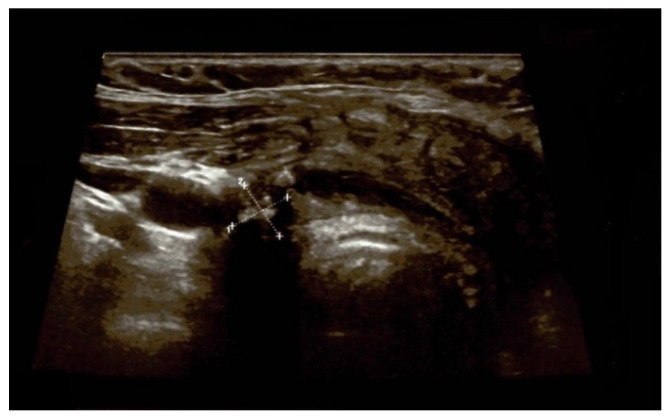
Anterior neck ultrasound after redo thyroidectomy: no thyroid remnant, and persistent post-operatory oedema.

**Table 1 jcm-13-05347-t001:** Studies addressing redo thyroidectomy according to our methods (the display follows the results section) [34,37,42,43,47,48,49,59,66,68] (abbreviations: N—number of patients; TORT—trans-oral robotic thyroidectomy).

Reference Number	First Author	Study Design	Studied Population	Outcome
[34]	Velayutham, P	nationwide survey (India)	101 surgeons (respondents)	45.5% of the respondents used intra-operative nerve monitoring in redo thyroidectomy
[37]	Haddadin, SW	retrospective study	255 patients with thyroid surgery	rate of unilateral recurrent laryngeal nerve injury as follows: ⏵ Total cohort: 6.7%. ⏵ Total/near-total versus bilateral/unilateral subtotal thyroidectomy: 25.9% versus 7.9%, *p* < 0.05). ⏵ Malignant versus benign conditions: 17.1% versus 4.7%, *p* < 0.05. ⏵ Males versus females: 12.2% versus 8.8%, *p* < 0.05. ⏵ Redo thyroidectomy: 8.3%
[42]	Dip, F	intercontinental, multidisciplinary, Delphi survey-based study	10 international experts in fluorescence imaging during endocrine (thyroid and parathyroid) surgery	indocyanine green is particularly important in redo surgery
[43]	Dalcı, K	retrospective study	25 patients with redo thyroidectomy (20/25) and parathyroidectomy (5/25)	radionuclide occult lesion localisation offered mean operative time of 100.6 min thyroglobulin decrease (*p* < 0.05)
[47]	Edafe, O	retrospective study	1913 patients who underwent thyroid and parathyroid surgery	1.3% = risk of redo operation due to post-operatory bleeding
[48]	Abboud, B	retrospective study	4153 patients with redo thyroidectomy	haematoma rate: 4% following redo completion thyroidectomy or recurrent thyroid disease
[49]	Doran, HE	retrospective study	52838 patients from UK Registry of Endocrine and Thyroid Surgery	1.2% rate of redo intervention for post-operatory bleeding
[59]	Oh, MY	retrospective study	10 patients who underwent redo TORT	Initial versus redo TORT: ⏵ Similar flat dissection time (*p* = 0.125). ⏵ Higher total operation time (*p* = 0.021). higher console time (*p* = 0.019)
[66]	Kim, DG	retrospective study	65 patients with trans-axillary robotic approach (redo) for recurrent thyroid cancer: ⏵ Completion total thyroidectomy (N1 = 26). ⏵ Completion total thyroidectomy + associated with modified radical neck node dissection (N2 = 16). ⏵ Modified radical neck node dissection (N3 = 23).	⏵ Rate of transitory hypocalcaemia: 26.2%. ⏵ Rate of permanent hypocalcaemia: 4.6%. ⏵ Rate of recurrent laryngeal nerve injury: 4.6%.
[68]	Doval, AF	cost analysis	22,521 patients who underwent thyroidectomy from National Surgical Quality Improvement Program database	median dollars per minute in primary versus redo thyroidectomy: $4.97 versus $8.12 (*p* < 0.0001)

## Data Availability

Other data are available on reasonable requests.

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
