# Peer review of "Redo Thyroidectomy: Updated Insights"

_jcm, 2024, doi:10.3390/jcm13185347_

Round 1

Reviewer 1 Report

Comments and Suggestions for Authors

Thank you for the possibility to review the manuscript titled: “Redo thyroidectomy: updated insights”. The study is interesting and provides and important overview of the topic. Thyroid surgery is a narrow branch of surgery that can be performed by general, endocrine, oncological surgeons or otolaryngologists and head & neck surgeons. Therefore, there is a constant discussion of this filed of surgery. The overall importance of the study is dictated by the number of specific complications such as recurrent laryngeal nerve paralysis and parathyroid dysfunction. Therefore, the aim of the current study is understandable and important for the day-to-day practice. Nevertheless, there are several matters that should be addressed.

-Authors state: “previously unsuspected carcinoma might be detected and, hence, it indicates a re-intervention as mandatory”. This sentence is misleading. Thyroid carcinoma does not always require thyroidectomy and lymph node dissection. This topic is debatable but according to the current guidelines only high-risk thyroid cancer according to ATA requires thyroidectomy and radioiodine treatment. There are several molecular types such as BRAF (+) that may also require thyroidectomy as a treatment of choice. Please consult the indicated manuscripts and the latest guidelines (some of them are listed bellow).

1)Omry-Orbach G. Risk Stratification in Differentiated Thyroid Cancer: An Ongoing Process. Rambam Maimonides Med J. 2016 Jan 28;7(1):e0003. doi: 10.5041/RMMJ.10230. PMID: 26886959; PMCID: PMC4737509.

2)Haugen BR, et al. 2015 American Thyroid Association Management Guidelines for Adult Patients with Thyroid Nodules and Differentiated Thyroid Cancer: The American Thyroid Association Guidelines Task Force on Thyroid Nodules and Differentiated Thyroid Cancer. Thyroid. 2016 Jan;26(1):1-133. doi: 10.1089/thy.2015.0020. PMID: 26462967; PMCID: PMC4739132.

-“Routine application of the intra-operative nerve monitoring amid thyroidectomy still represents a challenging issue across daily surgical practice, and most modern data showed a heterogeneous use of this technique all over the world with large variations between centres”. I do agree with the statement of heterogeneous use. However, it should be mentioned that most of the studies indicate a significant decrease in nerve damage with the use of neuromonitor.

-Authors state ”Yet, an immobile focal cord (but with the preservation of its anatomical integrity) helps keeping the voice [37].” This is also misleading the anatomically intact vocal cords do not produce adequate sound in case of recurrent laryngeal nerve damage. However, this discussion requires more data as in more than 90% of cases the nerve will improve its function if it was not cut during surgery and trauma is related to other factors.

“In most cases, recurrent laryngeal nerve paresis is transient, because 94.6% of patients have a complete restoration of the voice” as stated by Chiang FY, et al. Recurrent Laryngeal Nerve Palsy After Thyroidectomy With Routine Identification of the Recurrent Laryngeal NerveSurgery (2005) 137(3):342–7. doi: 10.1016/j.surg.2004.09.008

-Fluorescence-imaging techniques are not limited to indocyane green which is a relatively new area. There are other ways to identify parathyroid glands such as “stress-test”, 5-aminolevulenic acid etc. Some of these methods are discussed in the manuscript presented bellow:

Dolidze D, et al (2023) Prophylaxis of postoperative hypoparathyroidism in thyroid surgery. Folia Medica 65(2): 207-214. https://doi.org/10.3897/folmed.65.e75427

Subtotal thyroidectomy is an old and debatable procedure. According to the majority of guidelines a minimal volume of surgery is a lobectomy. Please change this in the manuscript.

-As the authods discuss Redo TORT (trans-oral robotic thyroidectomy) they should also mention endoscopic (non-robotic) procedures in the manuscript.

-Please change “She felt good” to “there were no complains”.

-Please change the term “totalization procedure”.

-A discussion of the current complications of thyroid surgery can be found in a review: “A Narrative Review of Preventive Central Lymph Node Dissection in Patients With Papillary Thyroid Cancer - A Necessity or an Excess. Front Oncol. 2022 Jun 29;12:906695. doi: 10.3389/fonc.2022.906695. PMID: 35847927; PMCID: PMC9278848.” You can find some potentially useful citations in the manuscript.

-There are type mistakes in the manuscript. Please review the language of the manuscript.

Please take into consideration the recommendation in the spirit of improving the quality of the submission.

Comments on the Quality of English Language

Moderate editing of English language required.

Author Response

Response to Review 1 Comments

Dear Reviewer,

Thank you very much for your time and your effort to review our manuscript.

We are very grateful for providing your valuable feedback on the article.

Here is our response and related amendment that has been made in the manuscript according to your review (marked in yellow color).

Thank you for the possibility to review the manuscript titled: “Redo thyroidectomy: updated insights”. The study is interesting and provides and important overview of the topic. Thyroid surgery is a narrow branch of surgery that can be performed by general, endocrine, oncological surgeons or otolaryngologists and head & neck surgeons. Therefore, there is a constant discussion of this filed of surgery. The overall importance of the study is dictated by the number of specific complications such as recurrent laryngeal nerve paralysis and parathyroid dysfunction. Therefore, the aim of the current study is understandable and important for the day-to-day practice. Nevertheless, there are several matters that should be addressed.

Authors state: “previously unsuspected carcinoma might be detected and, hence, it indicates a re-intervention as mandatory”. This sentence is misleading. Thyroid carcinoma does not always require thyroidectomy and lymph node dissection. This topic is debatable but according to the current guidelines only high-risk thyroid cancer according to ATA requires thyroidectomy and radioiodine treatment. There are several molecular types such as BRAF (+) that may also require thyroidectomy as a treatment of choice. Please consult the indicated manuscripts and the latest guidelines (some of them are listed bellow).

1)Omry-Orbach G. Risk Stratification in Differentiated Thyroid Cancer: An Ongoing Process. Rambam Maimonides Med J. 2016 Jan 28;7(1):e0003. doi: 10.5041/RMMJ.10230. PMID: 26886959; PMCID: PMC4737509.

2)Haugen BR, et al. 2015 American Thyroid Association Management Guidelines for Adult Patients with Thyroid Nodules and Differentiated Thyroid Cancer: The American Thyroid Association Guidelines Task Force on Thyroid Nodules and Differentiated Thyroid Cancer. Thyroid. 2016 Jan;26(1):1-133. doi:10.1089/thy.2015.0020. PMID: 26462967; PMCID: PMC4739132.

Thank you very much. We followed your recommendation. Thank you

“Routine application of the intra-operative nerve monitoring amid thyroidectomy still represents a challenging issue across daily surgical practice, and most modern data showed a heterogeneous use of this technique all over the world with large variations between centres”. I do agree with the statement of heterogeneous use. However, it should be mentioned that most of the studies indicate a significant decrease in nerve damage with the use of neuromonitor. Authors state ”Yet, an immobile focal cord (but with the preservation of its anatomical integrity) helps keeping the voice [37].” This is also misleading the anatomically intact vocal cords do not produce adequate sound in case of recurrent laryngeal nerve damage. However, this discussion requires more data as in more than 90% of cases the nerve will improve its function if it was not cut during surgery and trauma is related to other factors.

“In most cases, recurrent laryngeal nerve paresis is transient, because 94.6% of patients have a complete restoration of the voice” as stated by Chiang FY, et al. Recurrent Laryngeal Nerve Palsy After Thyroidectomy With Routine Identification of the Recurrent Laryngeal Nerve. Surgery (2005) 137(3):342–7. doi: 10.1016/j.surg.2004.09.008

Thank you very much. We mentioned this aspect based on your recommendation. Thank you

Fluorescence-imaging techniques are not limited to indocyane green which is a relatively new area. There are other ways to identify parathyroid glands such as “stress-test”, 5-aminolevulenic acid etc. Some of these methods are discussed in the manuscript presented bellow:

Dolidze D, et al (2023) Prophylaxis of postoperative hypoparathyroidism in thyroid surgery. Folia Medica 65(2): 207-214. https://doi.org/10.3897/folmed.65.e75427

Thank you very much. We followed your useful and interesting observation and pinpointed this specific aspect. Thank you

Subtotal thyroidectomy is an old and debatable procedure. According to the majority of guidelines a minimal volume of surgery is a lobectomy. Please change this in the manuscript.

Thank you very much. We corrected it all over the paper with 3 exceptions:

  1. One exception is a cited paper according to our search (we kept the original term of the original authors)

Haddadin SW, Mahasna AM, Abumekhleb IA, Almaaitah FS, Alhyari FMA, Alsaidat YM, Alkhataleen AA, Albaddawi YY, Alshehabat LAM, Makhamreh OH. Comparison of Recurrent Laryngeal Nerve Insult Incidence Post Thyroidectomy for Benign and Malignant Lesions. Med Arch. 2023;77(3):213-217. doi:10.5455/medarh.2023.77.213-217.

  1. Another is a cited paper as follows:

Cirocchi R, Trastulli S, Randolph J, Guarino S, Di Rocco G, Arezzo A, D'Andrea V, Santoro A, Barczyñski M, Avenia N. Total or near-total thyroidectomy versus subtotal thyroidectomy for multinodular non-toxic goitre in adults. Cochrane Database Syst Rev. 2015;2015(8):CD010370. doi:10.1002/14651858.CD010370.pub2.

  1. And the third is the original report of the case on point according to the patient’ initial record that was done by her first surgeon and we need to reproduce the facts (this is real life medicine)

Thank you very much

As the authods discuss Redo TORT (trans-oral robotic thyroidectomy) they should also mention endoscopic (non-robotic) procedures in the manuscript.

Thank you very much. We respectfully mention that according to our search and methods we found no other data than those already mentioned. Thank you

Please change “She felt good” to “there were no complains”.

Thank you very much. We corrected it.

Please change the term “totalization procedure”.

Thank you very much. We corrected it.

A discussion of the current complications of thyroid surgery can be found in a review: “A Narrative Review of Preventive Central Lymph Node Dissection in Patients With Papillary Thyroid Cancer - A Necessity or an Excess. Front Oncol. 2022 Jun 29;12:906695. doi: 10.3389/fonc.2022.906695. PMID: 35847927; PMCID: PMC9278848.” You can find some potentially useful citations in the manuscript.

Thank you very much. We followed your recommendation. Thank you

There are type mistakes in the manuscript. Please review the language of the manuscript.

Thank you very much. We corrected them. Thank you

Please take into consideration the recommendation in the spirit of improving the quality of the submission.

Thank you very much

Comments on the Quality of English Language: Moderate editing of English language required.

Thank you very much. We corrected it.

Thank you very much.

Reviewer 2 Report

Comments and Suggestions for Authors

This manuscript reviews the most recent data regarding redo thyroidectomy providing a comprehensive overview with real-world implications. However, the authors should add a discussion regarding the future directions of research in redo thyroidectomy and a figure resuming the different surgical approaches. Also, the introduction section should be expanded with more recent references regarding the epidemiology and diagnostic approaches of thyroid neoplasms (10.1158/1055-9965.EPI-21-1440; doi: 10.1530/ETJ-22-0146; 10.3389/fendo.2023.1101410; https://doi.org/10.1002/cncy.22224). Lastly, a spell and punctuation check should be performed.

Comments on the Quality of English Language

Minor English editing required

Author Response

Response to Review 2 Comments

Dear Reviewer,

Thank you very much for your time and your effort to review our manuscript.

We are very grateful for your insightful comments and observations, also, for providing your valuable feedback on the article.

Here is a point-by-point response and related amendments that have been made in the manuscript according to your review (marked in yellow color).

This manuscript reviews the most recent data regarding redo thyroidectomy providing a comprehensive overview with real-world implications.

Thank you very much.

However, the authors should add a discussion regarding the future directions of research in redo thyroidectomy and a figure resuming the different surgical approaches.

Thank you very much. We expanded this section at Discussion according to your recommendation. We respectfully mention that the analysis of the different types of the surgical approaches is out of our scope which is intended to be a brief update, but the approaches in redo thyroidectomy according to the current level of statistical evidence and amid our methods of search has been highlighted in Figure 1. Thank you

Also, the introduction section should be expanded with more recent references regarding the epidemiology and diagnostic approaches of thyroid neoplasms (10.1158/1055-9965.EPI-21-1440; doi:10.1530/ETJ-22-0146; 10.3389/fendo.2023.1101410; https://doi.org/10.1002/cncy.22224).

Thank you very much. We expanded the data according to your recommendation. Thank you

Lastly, a spell and punctuation check should be performed.

Thank you very much. We corrected it.

Thank you very much

Reviewer 3 Report

Comments and Suggestions for Authors

I appreciate the chance to evaluate the manuscript titled "Redo thyroidectomy: updated insights".

The authors aimed to provide an up-to-date review of the literature on redo thyroid surgery, supplemented by an extensive description of a case from their own clinical experience. Given the frequency of re-operative thyroid surgery in modern times, it is crucial to address the associated risks in a clear and scientific manner to prevent potential serious consequences.

The authors conducted a search for articles written in English using the terms "redo", "thyroid", "thyroidectomy", and "thyroid surgery" in the PubMed database. Only original studies published between January 1, 2020, and July 20, 2024, were included for further research. This is a well-thought-out strategy that constitutes the strength of this manuscript.

The introduction effectively acquaints the reader with the topics addressed throughout the remainder of the manuscript. The devised approach for analyzing PubMed data, as previously noted, is meticulously planned and precise. The case description and discussion are pertinent. The presented conclusions are consistent with the content of the article. In this case, I have no objections to the methodology.

The manuscript may not add much to the existing knowledge about redo thyroid surgery, but it organizes the existing data. As a result, readers can access all the essential and current information from a single source. Additionally, they can compare their own clinical experiences with those presented by the authors through the provided case description.

The number of self-citations in the manuscript is my primary concern. While this is not inherently problematic, the use of self-citations with footnotes to numbers 16 and 108 appears either unjustified in the text or necessitates further elaboration on the topic. I request that the authors consider expanding the explanation or deleting these references when they are editing the article.

Author Response

Response to Review 3 Comments

Dear Reviewer,

Thank you very much for your time and your effort to review our manuscript.

We are very grateful for your insightful comments and observations, also, for providing your valuable feedback on the article.

Here is a point-by-point response and related amendments that have been made in the manuscript according to your review (marked in yellow color).

I appreciate the chance to evaluate the manuscript titled "Redo thyroidectomy: updated insights".
The authors aimed to provide an up-to-date review of the literature on redo thyroid surgery, supplemented by an extensive description of a case from their own clinical experience. Given the frequency of re-operative thyroid surgery in modern times, it is crucial to address the associated risks in a clear and scientific manner to prevent potential serious consequences.
The authors conducted a search for articles written in English using the terms "redo", "thyroid", "thyroidectomy", and "thyroid surgery" in the PubMed database. Only original studies published between January 1, 2020, and July 20, 2024, were included for further research. This is a well-thought-out strategy that constitutes the strength of this manuscript.
Thank you very much. We really appreciate it!

The introduction effectively acquaints the reader with the topics addressed throughout the remainder of the manuscript. The devised approach for analyzing PubMed data, as previously noted, is meticulously planned and precise. The case description and discussion are pertinent. The presented conclusions are consistent with the content of the article. In this case, I have no objections to the methodology.
Thank you very much.

The manuscript may not add much to the existing knowledge about redo thyroid surgery, but it organizes the existing data. As a result, readers can access all the essential and current information from a single source. Additionally, they can compare their own clinical experiences with those presented by the authors through the provided case description.
Thank you very much.

The number of self-citations in the manuscript is my primary concern. While this is not inherently problematic, the use of self-citations with footnotes to numbers 16 and 108 appears either unjustified in the text or necessitates further elaboration on the topic. I request that the authors consider expanding the explanation or deleting these references when they are editing the article.

Thank you very much. We deleted them and expand the explanations in accordance to all recommendations.

Thank you very much
